# Psychological and Sociocultural Determinants in Childhood Asthma Disease: Impact on Quality of Life

**DOI:** 10.3390/ijerph19052652

**Published:** 2022-02-24

**Authors:** Sheila Plaza-González, María del Carmen Zabala-Baños, Álvaro Astasio-Picado, Jesús Jurado-Palomo

**Affiliations:** 1Nursing Department, Puerta de Hierro University Hospital, Majadahonda, 28222 Madrid, Spain; sheilapg22@hotmail.com; 2Physiotherapy, Nursing and Physiology Department, Faculty of Health Sciences, University of Castilla-La Mancha, 45600 Toledo, Spain; alvaro.astasio@uclm.es (Á.A.-P.); jesus.juradopalomo@uclm.es (J.J.-P.)

**Keywords:** bronchial asthma, pediatric patients, psychological factors, psychosomatic disease, personality character and emotions

## Abstract

Asthma is the most common chronic disease in childhood. The presence of this pathology in children leads to the appearance of different alterations (physical, psychological, social, etc.). Due to their high influence, the aim of this study is to understand these psychological and sociocultural determinants and their impact on the quality of life of asthmatic children. In order to determine the influence of these determinants on quality of life, a narrative review of 48 articles collected in different databases was carried out. Emotions are the most powerful precursor to producing an asthmatic attack. Anxiety and depression are the pathologies that appear frequently associated with childhood asthma, together with Attention-Deficit/Hyperactivity Disorder. In addition, the personality of these children seems to be characterized by shyness and impulsivity, although exceptionally it has been associated with psychopathic behaviors, aggressiveness, and cases of psychosis. School performance is impaired and bullying occurs more frequently. Likewise, dysfunctional family relationships and lower socioeconomic status have a negative impact on the severity and management of asthma. In short, the quality of life of asthmatic children is lower due to the presence of the aforementioned psychological and sociocultural determinants.

## 1. Introduction

Asthma is the most common chronic disease in children. The main risk factors have been identified as environmental exposure to certain particles and substances (also known as precursors) combined with a genetic predisposition, leading to airway irritation or allergic reactions. They are subdivided into two types: allergic precursors and non-allergic precursors [1].

Allergic precursors refer to substances such as household allergens, pollen, and exposure to smoke and non-allergic precursors refer to phenomena such as cold air, physical exercise, or intense emotions such as fear, anger and sadness. According to the WHO [1], non-allergic precursors are more difficult to avoid and treat, as they are unpredictable; an example is when a person experiences stressful situations, weather changes, and intense emotions, as well as crying or laughing. These precursors trigger the first symptoms of asthma by activating certain brain centers that control emotions and socialization, so special attention must be paid to the symptomatology in order to reduce the severity of the attack as quickly as possible.

Both precursors favor the onset of hyperventilation, hypercapnia, and bronchoconstriction through symptoms such as coughing, wheezing, chest tightness, and shortness of breath, leading the patient to experience a very unpleasant sensation of breathlessness.

When these symptoms develop chronically, children may experience daytime fatigue, insomnia, and increased school absenteeism. 

In addition, studies have shown that asthmatic children have a genetic vulnerability to behavioral disturbances, leading to chronification of symptoms, increasing the frequency of asthmatic attacks and sleep pattern disturbances [2]. 

Another factor to note is that it has been shown that obesity and overweight in children with asthma lead to impaired immunity due to vitamin D deficiency, increasing the risk of viral infections with reduced responses to inhaled corticosteroids, substances that form a large part of the treatment of pediatric asthma patients, so it is essential to introduce a healthy lifestyle in this type of patient [3,4].

Beyond the physiological consequences of asthma, children may experience alterations in personality traits, psychological and sociocultural factors, and quality of life.

Psychologists and specialists in the field of psychosomatic diseases point to asthma as a major factor in the development of emotional, social, and economic difficulties [1]. However, other professionals take a completely opposite view, arguing that it is the child’s unexpressed internalizing and externalizing factors that lead to the onset of asthma.

The externalizing factor is characterized by outwardly directed behaviors affecting and involving other people, sometimes deviating from the established norms of society. This syndrome refers to inattention, aggressiveness, delinquent behavior, problems expressing conflict with other people, and disobedience.

On the other hand, the internalizing factor reflects in the child non-adaptive behaviors that produce self-harm, also involving emotional disturbances such as depression, social withdrawal, anxiety, and somatic complaints without medical justification.

According to the WHO, asthma is the symbolic expression of unconscious conflicts and repressed desires of the child that lead to pure anxiety, depression, poor school performance, affective problems, attention disorders, ADHD, sleep problems, and problems of internalization and externalization [5].

Some theories have shown that emotional and sociocultural factors can maintain or cause childhood asthmatic pathology. For this reason, it has become necessary to understand these factors and determine their dimensions and influence on the clinical setting and evolution of the pathology, as well as on the patient’s quality of life.

The main objective is to understand the influence of psychological and sociocultural determinants on the quality of life of asthmatic children.

However, the secondary objectives are to identify the most frequent psychological and physiological alterations associated with childhood asthma, study whether the mental health of asthmatic children is further compromised by school bullying and whether this problem influences the control of the asthmatic disease, to determine whether lower socioeconomic status severely compromises the health of children suffering from this psychosomatically based disease, to Determine whether there is an association between poor academic performance in children with asthma and poorer asthma management, and to explore whether functional and dysfunctional family and social relationships modify the child’s emotional and physiological response to chronic asthmatic disease.

## 2. Methodology

In order to answer these objectives, an exhaustive literature search was carried out. The search string selected was based on the keywords “bronchial asthma”, “asthma”, “child”, “pediatric patients”, “kids”, “psychological factors”, “psychological disorders”, “psychosomatic disease”, “personality” “character”, and “emotions” which were combined with the Boolean operators AND and OR as follows: (“bronchial asthma” OR “asthma”) AND (“child*” OR “pediatric patients” OR “kids”) AND (“psychological factors” OR “psychological disorder*” OR “psychosomatic disease*”OR “personality” OR “character” OR “emotion*”).

This search string was entered into the following databases: Pubmed, Web of Science, PsicoDoc, Pubpsych, Scopus, PsycInfo, Psychology and Behavioral Sciences Sollections, Psychology Database (Proquest Central), Cochrane, and Cinahl in order to obtain information relating childhood asthma (“bronchial asthma” OR “asthma”) AND (“child*” OR “pediatric patients” OR “kids”) to different psychological factors, psychological factors and/or emotions (“psychological factors” OR “psychological disorder*” OR “psychosomatic disease*” OR “personality” OR “character” OR “emotion*”) that this disease with a psychosomatic component causes. The articles selected for this research were chosen on the basis of a number of inclusion and exclusion criteria (Table 1).

To carry out the article selection process, the search string was entered into the different databases, gradually applying the above inclusion and exclusion criteria (Table 2; Figure 1).

## 3. Results

### 3.1. Psychological Dimension

#### 3.1.1. Anxiety and Depression

Maria C Mirabelli et al. [6] state that the stress generated by asthma comorbidities increases the number of asthmatic attacks and the number of visits to the emergency department.

In addition, findings described in other articles [7] confirm the previous hypothesis, pointing out that the greater the severity of asthma, the more frequent the need for health services and the higher the absenteeism from school.

In the same dynamic as the previous articles, Lene Hammer-Helmich et al. [8,9] mention that phobias, depression, and anxiety secondary to allergic diseases such as asthma produce changes in children’s behavior.

In another context, other authors [10] determined that if the asthmatic child suffers from obesity, his or her probability of suffering from anxiety, affective, or psychiatric or behavioral disorders will be multiplied.

Deniz Aktan et al. [11] found that the IgE cells expressed during asthma attacks could be responsible for subsequent mental health problems (anxiety, depression).

Lilly Shanahan’s study [12] highlights the physiological reason why the combination of asthma and depression worsens asthmatic children.

This combination increases CRP and NR3CH values, which in turn increases inflammation of bodily organs such as the lungs.

However, other research [13] confirms that subjects’ adherence to treatment is highly dependent on the level of stress and depression experienced by these children.

#### 3.1.2. ADHD

The findings of Ribera Asensio [14] state that asthma is an added stressor for the child and this factor conditions the appearance of biopsychosocial changes and emotional and behavioral repercussions.

In another context, other authors determined that children with obesity have a greater tendency to develop anxiety and affective, behavioral, and psychiatric disorders, which are directly related to the onset of asthma [14,15].

From another perspective, the paper by authors Marcia Winter et al. [15,16,17,18] show that the severity of the asthmatic process exacerbates both the expression of anxiety and depression as well as irritability and impulsivity, which pose a risk to the child’s personal development and safety.

In this context, some authors [4] confirmed that the presence of depression in asthmatic children reduces adherence to treatment leading to poorer control of the disease and triggering an increased number of comorbidities and death.

With regard to Attention-Deficit/Hyperactivity Disorder, Maya [16] pointed out that asthmatic children are prone to ADHD.

Other authors [17] state that asthma is a risk factor for developing ADHD, mainly in children with anxiety and depression who do not have adequate control of the disease.

#### 3.1.3. Personality Traits

Thanks to the paper conducted by Dhristie Bhagat [19], it has been shown that asthmatic children are more shy and perform well academically.

In addition, Sara Agnafors together with other authors [20] managed to relate psychopathic behaviors (behaviors expressed by people with a cold, impulsive, aggressive, and manipulative character who violate social norms) and psychosis (abnormal perceptions and ideas of reality whose main symptoms are delusions and hallucinations with substance abuse during childhood and adolescence) with an increased risk of experiencing asthma in these same life periods [21].

In another study by Tatiana Lacruz-Gascón et al. [22], low neuroticism and extroversion are personality traits that favor emotional competence and control in children with asthma. Another aspect that has aroused interest in relation to childhood asthma is the level of stress in asthmatic children and the emotions generated by suffering from this chronic disease.

#### 3.1.4. Emotions and Stress

Some authors [23] argue that non-allergic precursors such as emotions (laughing, crying, etc.) or environmental factors (rainfall, etc.) are the most difficult to control and prevent, as they are unpredictable.

As in the previous study, Agnafors and Norman Kjellström [24] support the theory that emotions are the most dangerous trigger of asthma attacks in children, notably affecting the way they breathe. 

On the other hand, authors Sofia Edvinsson et al. [25] state that the concomitance of asthma and allergy poses a risk to mental health due to increased emotional symptomatology.

In the same line of research, Erin Rodriguez et al. [26] concluded that better control of this pathology and the individual’s commitment to following treatment lead to a reduction in stress in childhood asthma.

Other authors [27], however, determined that the stress produced during asthmatic disease leads to a chronification of anxiety.

Furthermore, as stated by Erin Rodriguez et al. [28], stress leads to poor disease control in asthmatic children, leading to emotional and behavioral problems. Positive thinking together with symptom distraction reduces emotional and behavioral problems.

Finally, the findings discovered by Catalina Bronstein [29] determine that childhood asthma involves a sense of emotional emptiness conditioned by anxiety in these patients. These statements are what led the author to consider and define an asthmatic attack as a feeling that cannot be expressed and that causes physical and emotional discomfort to the subject.

A recent article [30] shows that the inflammatory regulator NR3C1 is closely linked to emotions, stress, and socioeconomic status. Deficiency of NR3C1 causes cortisol to act in an uncontrolled manner leading to chronic asthma.

Finally, in relation to this contribution, Thomas Ritz et al. [31] explained that the vagal stimulation produced by induced or provoked emotions causes the airways to swell and prevents the child from breathing normally.

### 3.2. Sociocultural Dimension

#### 3.2.1. Bullying

In a study conducted by Mary Stephens et al. [32], it was found that children with asthma are more likely to be bullied. In addition, bullying influences the development of stress or depression, leading to a decrease in academic performance and a higher prevalence of truancy.

However, other authors [33,34] also report that asthmatic children living with high physical limitations, inability to have pets, preoccupation with asthma, medication management, and eating restrictions are more often lonely and discriminated against.

#### 3.2.2. Academic Performance

Some authors [35] confirmed that childhood asthma results in a higher number of emergency room visits and missed school days. For this reason, asthma exacerbations alter children’s sleep patterns.

Richard Layte’s publication [34] supports that pediatric asthma patients have poorer academic outcomes accompanied by lower cognitive development and less willingness to learn.

However, other authors [17] argued that childhood asthma is not associated with poor academic performance or behavioral problems.

#### 3.2.3. Socioeconomic Status

As far as socioeconomic status is concerned, as stated by Maya Nanda et al. [16,17], lower socioeconomic status favors the development of atopic diseases such as asthma.

The authors Edith Chen et al. [36] in their publication argue that having a low socioeconomic status increases mortality, morbidity, and emergency and hospital admissions in asthmatic children.

In addition to the above, other authors [37] confirmed that ethnic minorities are a group highly affected by asthma due to the stressors they experience.

Finally, Ledina Imami et al. [38] found that lower socioeconomic status is associated with difficulties in asthma management and with impulsivity, disobedience, anxiety, and helplessness in patients.

#### 3.2.4. Family and Social Relationships

Regarding family and social relationships, some authors [38] highlighted that maternal education and caring maternal behaviors towards an asthmatic child lead to a decrease in internalizing problems and an enhancement of immune functioning.

According to Shelley Boeschoten et al. [39] complications of the disease such as ICU admission affect patients and relatives psychologically and psychiatrically, as parents may develop post-traumatic stress syndrome.

In another context, a groundbreaking study [40] established that sibling relationships have benefits for children with asthma as they produce a moderating effect by boosting self-esteem, improving interpersonal and environmental relationships, increasing emotional control, and optimizing body image management in these children.

The authors JungHa Lim et al. noted that if parents of asthmatic children suffer from depression, there is an increase in internalizing responses by these children [41].

Marcia Winter et al. [15] argued that asthma affects feelings of security in children with asthma.

Finally, in another study [42], it was observed that the feelings and emotions expressed to a greater extent by parents (anger, uncertainty, fear, depression, guilt, or helplessness) are also expressed by children.

### 3.3. Quality of Life

The quality of life of asthmatic children is influenced by a multitude of factors, both psychological and physical. For this reason, many authors have focused on investigating the impact of these factors and what the best resources are to enhance this quality of life.

Authors Hannah Kansen et al. [23] found that the quality of life of pediatric asthma patients is influenced by two types of precursors: allergic and non-allergic. In this study, allergic triggers were found to be easier to manage and control and therefore would not lead to such a high reduction in quality of life.

In order to control emotionally triggered hyperventilations, Garrt Connett [24] proposed breathing techniques and their management as tools to improve the quality of life of these patients.

To complement these tools, research by Kimberly Raymond [43] presented six strategies that enhance the quality of life of these users, the most effective of which is the avoidance of triggers.

In another context, other authors [44] concluded that the concomitance of anxiety and asthma in children, by increasing the amount of medication and the frequency of symptoms, makes quality of life worse.

However, if children have the optimal ability to control and differentiate their emotions and pay reduced attention to possible precursors of asthmatic attacks, they will have a better quality of life, as stated by Lahaye et al. [45].

Some authors [46] reflected that the quality of life of asthmatic children is lower than that of healthy children due to the presence of allergies. According to a recent study [47], this low quality of life causes a detriment in school performance in asthmatic children due to the presence of psychological and physiological factors, with psychological factors often being responsible for the appearance of physiological ones.

Similarly, other authors [48] highlight the role of parents, pointing out that if parents, due to their overprotection, make their child fragile, their quality of life will eventually be damaged, as they will not have achieved the necessary skills to cope with their illness on their own.

Authors Moola et al. [49] argue that the increased incidence of anxiety, depression, behavioral disorders, post-traumatic stress disorder, absenteeism, and school disruption decreases the quality of life of asthmatics because these factors lead to increased feelings of frustration and hopelessness.

However, other authors [50] found that, if disease control is poor, the asthmatic child’s biopsychosocial domain will be significantly affected and therefore their quality of life will be affected as well.

Additionally, among Thabrew’s findings [51], it was highlighted that cyber-interventions are not effective in improving the quality of life of asthmatic children and that new technological tools need to be developed.

Finally, the paper by Magali Lahaye et al. [52] found that high scores on extroversion and benevolence together with low scores on neuroticism were associated with better quality of life in asthmatic children (Table 1).

## 4. Discussion

As the aim of this research was to identify and determine the influence of psychological and sociocultural determinants on the quality of life of asthmatic children, the results show that this psychosomatic disease is closely linked to the suffering of anxiety, Tocs, and depression in children.

Similarly, there are numerous indications that affirm its relationship with Attention-Deficit/Hyperactivity Disorder, impulsivity, and behavioral problems. This association is subject to the fact that in asthmatic children with added stressors during the course of the disease, the modulation of behavior or emotions is altered, thus affecting emotional control and the regulation of impulsivity and ADHD [12,28,40].

Personality is a variable that influences symptom control, adherence to treatment, quality of life, and the duration and severity of asthma. Children with ADHD have been shown to be shy, insecure, and impulsive, have a low frustration tolerance, and a low level of perseveration due to fatigue. This compendium of characteristics makes it difficult for the patient to adhere to treatment, which in turn leads to an increase in symptoms and their severity and therefore to a poorer quality of life [4,6,21].

However, some personality traits such as low neuroticism or high extroversion favor emotional control in asthmatic children [2,28].

It should be noted that psychopathic behavior, episodes of psychosis, or substance abuse increase the risk of asthmatic crises in these children [16,27].

Numerous studies point to emotions as the most unpredictable and dangerous precursor to asthmatic attacks. This is because allergic precursors can be avoided; however, experiencing a series of emotionally regulated processes is more difficult to control and prevent [21].

In fact, some authors define asthma as a feeling of emptiness that is triggered by a feeling that cannot be expressed, and which causes the individual physical and emotional discomfort [35,52].

Confidence is vital for this type of patient, as children who see themselves as having sufficient skills to stop the asthmatic attack improve their self-confidence and the resolution of the crisis in an adequate manner. For this reason, it is essential that patients know why an asthma attack occurs and the best way to resolve it. The physiological explanation as to why emotions sometimes cause asthmatic attacks lies in the fact that they produce a vagal stimulation in the body that causes the airways to become inflamed and prevents the child from breathing normally. In addition, some regulators of inflammation such as NR3C1 or cortisol are altered in the presence of stress or low socioeconomic status, causing breathing difficulties for the child [11,29,39,42].

It has been shown that asthma can affect sleep quality or obesity. Both phenomena have been linked to decreased immunity [19].

In addition, because the immune system plays a very important role in the development of psychosomatic diseases such as asthma, it should be taken into account that if the patient has mental health problems such as depression, this leads to immunodeficiency due to hyper-release of IgE, which aggravates asthma attacks [2,39].

However, it is not only the severity of asthma attacks that is influenced by physiological or psychological factors, but the sociocultural sphere of the patient also plays an important role [23].

One example is bullying; being an asthma sufferer makes you more vulnerable to bullying during the school period. This bullying is present in children who, because of asthma, cannot participate in sports activities, cannot have pets, need a lot of medication, or require dietary restrictions. Other authors postulate that this discrimination is based solely on socioeconomic factors [1,30,31,32].

From another perspective, there is controversy as to whether asthma conditions the presence of academic limitations. One hypothesis argues that asthma, by leading to emergency room visits and days of hospitalization, reduces children’s study time and performance, while other theories maintain that the appearance of sleep problems during the disease process, due to physical exhaustion, leads to alterations in school performance and psychological problems in the patient [1,45].

Children of lower socioeconomic status are more frequently diagnosed with asthma, and their asthma tends to be more severe and persistent. This is because in disadvantaged populations, inflammatory processes are exacerbated and asthma attacks occur more frequently. Another reason for this is the multitude of stressors experienced by ethnic minorities. Moreover, stressful events and negative feelings lead to airway inflammation and thus airflow obstruction [11,14,20].

This information leads to the conclusion that the chronicity of negative emotions in socioeconomic minorities generated by bullying and dysfunctional family relationships leads to a greater impact of asthmatic disease in children [10,13].

It has been shown that children’s psychological well-being and immune functioning is higher when mother–child relationships are very close and affectionate behaviors are consistently expressed. However, if these children’s immediate environment is psychologically or psychiatrically disturbed, emotional and physiological dysregulations appear in these children and make them worse. In the same way, the worsening of these children causes the parents to suffer anger, uncertainty, fear, guilt, or helplessness, thus forming a vicious feedback loop. Undoubtedly, sibling figures form a very powerful tool to calibrate this feedback loop, as healthy sibling relationships produce benefits for children with asthma, as they enhance self-esteem, security, and interpersonal relationships, providing the child with tools for adequate emotional control and improved body image management [35,37,45,46].

It is a fact that the quality of life of asthmatic children is lower than that of healthy children. One of the reasons for this is that these children are more susceptible to psychological and physiological comorbidities, and sometimes the psychological comorbidities explain the presence of physiological comorbidities. The presence of anxiety, depression, behavioral disorders, post-traumatic stress, absenteeism, and school interruption diminish the quality of life, as children feel a sense of frustration and hopelessness that prevents them from having adequate emotional control. However, in extroverted children with low levels of neuroticism, quality of life is increased [9,15,26].

Another reason why asthmatic children have a lower quality of life is based on the fact that in the presence of allergies, these children may mismanage the situation. For this reason, the quality of life of these children depends on the degree of control of the disease on their part; if the disease is under their control, the quality of life will be better. On the other hand, if the parents turn the child into what is known as a “fragile child” by preventing them from acquiring the competencies to cope with the disease on their own during the course of their life, their quality of life will be reduced [51,52].

To improve the quality of life in these patients, experts recommend physical activity that the patient can tolerate and proper management of emotions. Similarly, they state that the most effective strategy to prevent the asthmatic attack from occurring is “trigger avoidance”; if the precursor is avoided, the attack will not occur. [38,47]

## 5. Conclusions

With this review, we update the evidence to affirm that asthma is a disease with a psychosomatic basis and that the presence of negative psychological and sociocultural factors influences the quality of life of asthmatic children. Children with asthma are more often obese or overweight and have impaired immune systems and sleep quality. The health of children with asthma is further compromised if they are bullied or harassed at school. Asthmatic children generally perform worse academically and have lower socioeconomic status than healthy children. Dysfunctional family and social relationships in children with asthma negatively influence asthma management and quality of life.

## Figures and Tables

**Figure 1 ijerph-19-02652-f001:**
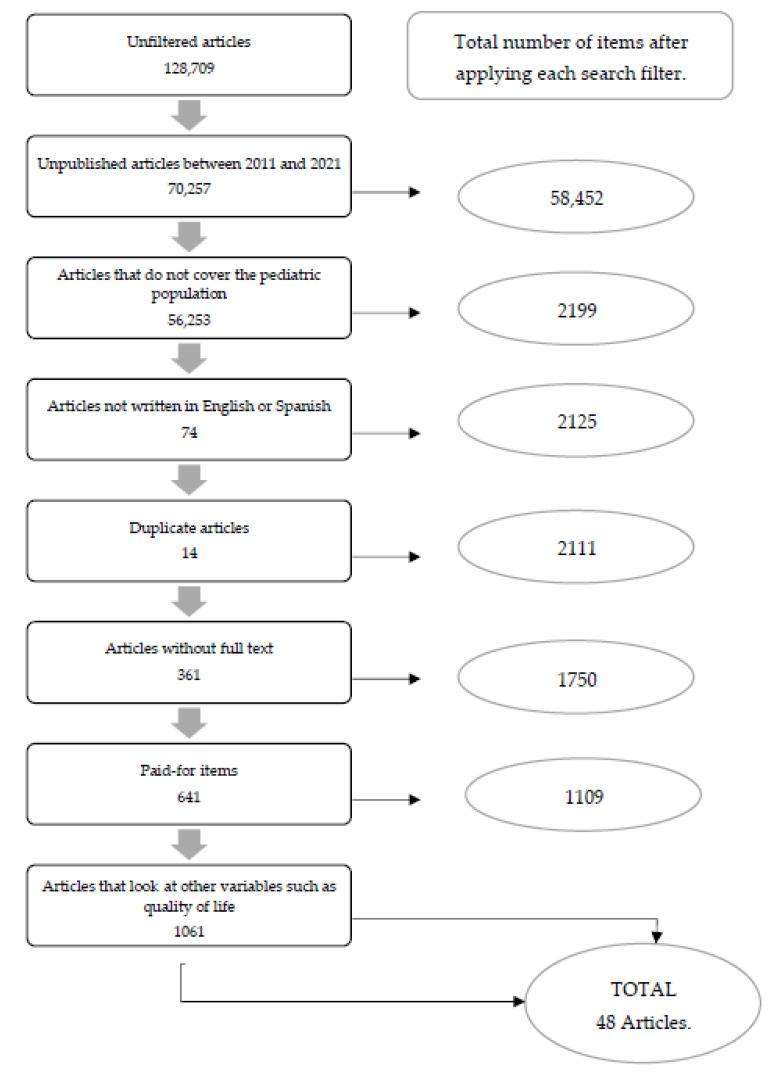
Flow diagram (prepared by the authors).

**Table 1 ijerph-19-02652-t001:** Criteria of inclusion and exclusion (own elaboration).

Inclusion Criteria		Exclusion Criteria
Articles covering information from the last 10 years (2011–2021)	Articles whose sample is a pediatric population	Articles whose study population is adults or the elderly
Articles in English or Spanish	Articles with full text	Documents containing only abstracts
Articles looking at other variables such as quality of life or personality		

**Table 2 ijerph-19-02652-t002:** Article selection process.

	Unfiltered Articles	Articles from 2011 to 2021	Articles Containing Pediatric Population (0–12 Years)	Articles in English and Spanish	Duplicate Articles	Full Text Articles	Free Articles	Total Articles That Fit the Topic under Study
PUBMED	983	326	216	206	203	193	84	7
WEB OF SCIENCE	1653	601	409	380	377	348	150	12
PSICODOC	16	6	1	1	1	1	1	1
PUBPSYCH	648	159	158	146	146	145	45	7
SCOPUS	1.528	540	382	371	371	369	166	8
SCIELO	0	0	0	0	0	0	0	0
PSYCINFO	908	228	144	139	134	39	5	1
PSYCHOLOGY AND BEHAVIORAL SCIENCES SOLLECTIONS	5.305	2.237	240	240	239	234	234	7
PSYCHOLOGY DATABASE (PROQUEST CENTRAL)	108.187	48.794	404	401	399	234	234	3
CUIDEN	0	0	0	0	0	0	0	0
COCHRANE	51	34	26	26	26	26	26	1
CINAHL	9.430	5.527	219	215	215	161	161	1
TOTAL	128.709	58.452	2.199	2.125	2.111	1.750	1.109	48

## Data Availability

Not applicable.

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
