# Peer review of "Psychological and Sociocultural Determinants in Childhood Asthma Disease: Impact on Quality of Life"

_ijerph, 2022, doi:10.3390/ijerph19052652_

Round 1

Reviewer 1 Report

The review examines research regarding childhood asthma and quality of life. As the authors mention, asthma is the most common chronic disease in childhood, lending importance to the review. The following are offered as points of clarity, to further strengthen the submission:

  1. The main concern with this very well-written review is the stated reliance on only “free access” content. The authors should clarify whether this means they did not/could not consult any academic peer-reviewed journals that are not all "open-access"/free without even an institutional subscription required? If so (only “open-access” reviewed), the authors should provide a limitation in that regard. Examining Table 1, it is clear that the dramatic drop in included studies (beyond the 2011 to 2021 requirement) is when the process moved from Full text to Free. This demonstrates a dramatic proportion of appropriate (at least potentially) studies were excluded due to lack of author access. How might this have affected the review?

Again, the review is succinct, well-written, and a useful summary of the state of empirical knowledge on the psychological and many sociocultural factors regarding asthma and quality of life in children. The access issue is my only concern.

Author Response

Dear reviewer,

We appreciate your assessment of the manuscript and we highly value major and minor suggestions to enrich it.

We appreciate your concern and we share it. The authors clarify that we access all paid journals through the institutional subscription of our reference center (Universidad de Castilla la Mancha). We have corrected the error of including the free access criterion in table 1, and after eliminating it we have verified that the results of this work are not modified.

We especially appreciate your excellent review of the article. We have proceeded to review it completely and in detail. We hope it is of your consideration.

Very thankful.

Reviewer 2 Report

Thank you for the opportunity to review this study entitled “Psychological and sociocultural determinants in childhood asthma disease: impact on quality of life.” (ijerph-1610411).

The manuscript presented a narrative review of 48 articles, exploring the role of emotional and socio-cultural factors in contributing to childhood asthma disease.

In my opinion the research topic is relevant, and the review is interesting. However, there are some issues that need to be addressed before the paper will be suitable for publication.

  • Abstract: according to the IJERPH’s guidelines, headings should be removed.
  • Abstract: please define more clearly and firmly the general objective of the review.
  • Introduction: “Asthma is the most common chronic disease in children”. Please provide a reference for this statement.
  • Lines 70-72. The authors claim to report Freud's thoughts by him, but they do not cite any of his texts by him. The in-text citation number five [5] is about other authors, who do not mention Freud in their paper. Please be more careful.
  • The introduction should be thorough, highlighting the need for this review and the contribution it can make.
  • Lines 78-91 should be moved at the end of the “Introduction” section.
  • Results: Please delete the initial of the author’s first name in the in-text citations, when they are unnecessary.
  • There are no in-text citations in the discussions. Instead, the "Discussion" section should present a discussion and expansion on the results presented. Therefore, this section should be revised entirely.
  • In the "Conclusions" section please explain the usefulness of this review and the practical implications that may derive from the conclusions reached by the authors.

Author Response

Dear reviewer,

We appreciate your assessment of the manuscript and greatly appreciate major and minor suggestions to enrich it.

We proceed to respond to your concerns:

1) Summary: Headings have been removed per IJERPH guidelines. Just as he defined it more clearly and concisely of the general objective of the study.

2) Introduction: The text is modified and justified with a new and updated bibliographic citation.

3) Lines 70-72. The argumentation is reviewed and modified according to the recommendations of the WHO.

4) The introduction is revised and the revision is highlighted according to the general objective of the study.

5) Lines 78-91 are moved to the end of the "Introduction" section.

6) Results: the initial of the author's name is eliminated in the citations in the text.

7) Discussion: due to a computer problem, the citations in the discussion section were not recorded. The discussion has been revised and updated.

8) Conclusions: the section is updated according to the conclusions reached by the authors.

We especially appreciate your excellent review of the article. We have proceeded to review it completely and in detail. We hope it is of your consideration.

Very thankful.

Round 2

Reviewer 2 Report

The authors have correctly resolved all the issues raised by my comments. Thank you